# Simultaneous Quantitative Analysis of Six Isothiazolinones in Water-Based Adhesive Used for Food Contact Materials by High-Performance Liquid Chromatography–Tandem Mass Spectrometry (HPLC–MS/MS)

**DOI:** 10.3390/molecules24213894

**Published:** 2019-10-29

**Authors:** Huaining Zhong, Zicheng Li, Sheng Chen, Ying Zeng, Jianguo Zheng, You Zeng, Dan Li

**Affiliations:** 1Guangdong Provincial Key Laboratory of import and export technical measures of animal, plant and food, Guangzhou 510623, China; marco_zhong@iqtc.cn (H.Z.); lizc@iqtc.cn (Z.L.); chens@iqtc.cn (S.C.); zengy@iqtc.cn (Y.Z.); lid@iqtc.cn (D.L.); 2Guangzhou Customs District Technology Center, Guangzhou 510623, China; 3Guangzhou Institute for Food Inspection, Guangzhou 511400, China; zengyou1225@163.com

**Keywords:** analysis, isothiazolinones, water-based adhesive, food contact materials, HPLC–MS/MS

## Abstract

In this study, a target analytical approach using high-performance liquid chromatography–tandem mass spectrometry (HPLC–MS/MS) was developed to simultaneously determine six isothiazolinones containing 2-Methylisothiazol-3(2H)-one (MI), 5-Chloro-2-methyl-4-isothiazolin-3-one (CMI), 1,2-benzisothiazolin-3-one (BIT), 2-Octyl-3(2H)-isothiazolinone (OIT), Dichlorooctylisothiazolinone (DCOIT), and 2-methyl-1,2-benzisothiazolin-3-one (MBIT) in water-based adhesive used for food contact materials. The main factors affecting extraction efficiency such as extraction method, extraction time, extraction solvent, and solid–liquid ratio have been evaluated by using real adhesive samples. Multiple-reaction monitoring (MRM) was used for the qualitative and quantitative analyses of targeted isothiazolinones. This method was demonstrated as an effective and reliable technique for detecting multiple isothiazolinones with satisfactory recoveries (81.5~107.3%), and the limits of detection (LOD) and quantification (LOQ) were obtained at a low level. This method was validated and applied to the determination of six isothiazolinones in commercial water-based adhesives. The present results revealed that these adhesives contained a combination of isothiazolinones (BIT, MI, CMI, and MBIT) with the concentration ranging from 2.27 to 123.5 mg/kg. To our knowledge, it is the first time it has been reported that MBIT was detected in water-based adhesives used for food contact materials, which requires a further investigation for its migration to food and the risk to human health.

## 1. Introduction

Isothiazolinones are a group of heterocyclic sulfur-containing compounds, which mainly include 2-Methylisothiazol-3(2H)-one (MI), 5-Chloro-2-methyl-4-chloroisothiazolinone-3-one (CMI), 1,2-Benzothiazol-3(2H)-one (BIT), and 2-Octyl-3(2H)-isothiazolinone (OIT), Dichlorooctylisothiazolinone (DCOIT), and 2-methyl-1,2-benzisothiazolin-3-one (MBIT), etc. (see Figure 1) [1,2]. Owing to their strong bactericide and fungicide properties, isothiazolinones are widely used as effective preservatives to control the growth of microbes in various applications, such as cosmetics [3], paint [4,5,6], detergents [7,8], adhesive [9,10], paper and cardboard [11,12], and consumer products [13,14,15].

Adhesive plays a very important role in food contact materials as it can provide a very good adhesion to various substrates, helping the packaging maintain the quality of the food and protect the food from contamination. More than 80% of food contact materials contains various kinds of adhesives [16], among which solvent-based adhesives and water-based adhesives are in predominant use. In recent years, water-based adhesive continues to attract attention as it is considered to be an environmentally friendly and efficient alternative to solvent-based adhesives owing to its low-volatile organic compounds (VOC), adjustable permeability, and economic practicality. However, one of main concern about the application of water-based adhesives in food packaging is that with high content of moisture contained compared to solvent-based adhesive, they are susceptible to attacks from bacteria and fungi. For this reason, the isothiazolinones are applied to water-based adhesives providing the protection against microorganisms with its broad-spectrum activity at low use levels.

Several studies have revealed that isothiazolinones including MI, BIT, OIT, and MCI could cause allergic contact dermatitis [1,2,3], new isothiazolinone derivatives such as Dichloro-octylisothiazolinone (DCOIT) and 2-methyl-1,2-benzisothiazolin-3-one (MBIT) were also found to be the potential source of allergy [17,18], attracting an increasing concern about their risk to human health. An epidemiological study conducted in Europe found that total of 6.0% among 3434 tested participants had positive patch test reactions to MI, suggesting that clinically relevant MI contact allergy remains prevalent across European countries [19]. Another study from a North America group during the period of 2013-2014 also indicated that 305 patients (6.3%) had a positive reaction to MCI/MI, which is a significant increase from the previous cycle (5.0%, 2011–2012) [20]. The cosmetics and household products are believed to represent the main source of exposure of certain kinds of isothiazolinones according to literature search. However, few studies have examined the exposure contribution made by food contact materials, even though isothiazolinones are frequently applied to food contact water-based adhesives. Moreover, the information for toxicity of isothiazolinones is very limited as most studies were carried out more than 20 years previously [21]. Consequently, it is difficult to perform a risk assessment on isothiazolinones originating from food contact materials, and few specific legislative measures have been established to response the challenge. In EU and China, given that the absence of specific regulation for food contact adhesives, the compliance check for isothiazolinones used for food contact water-based adhesive shall subject to framework regulation EU 1935-2004 [22] and China GB 4806.1-2016 [23] guarantying that food contact materials shall “do not transfer their constituents to food in quantities which could endanger human health”. To establish the appropriate regulation and to ensure the safe use of isothiazolinones in food contact materials, it is important to conduct more studies to obtain sufficient toxicity and exposure data for further risk assessment.

Several analytical techniques have been developed for the determination of isothiazolinones in various matrices, which mainly refer to liquid chromatography (LC) [24,25], gas chromatography-mass spectrometry (GC-MS) [26,27], and liquid chromatography–tandem mass spectrometry (LC–MS/MS) [28,29,30,31]. LC–MS/MS are considered to be a powerful technique with the advantages of high selectivity, sensitivity, and quantitative property for non-volatile compounds. However, most of the analysis techniques applied focus mainly on personal products and environmental matrices, such as cosmetics, paint, and water, etc., and the target analytes are limited to MI, CMI, BIT, and OIT. Considering that DCOIT and MBIT are applied in food contact adhesives, and limited information available for the occurrence of isothiazolinones in food contact materials, it is, therefore, a urgent demand to develop an accurate and sensitive analytical method that is suitable to quantify multiple isothiazolinones in water-based adhesive.

The main objective of this study is to establish an analytical method that is suitable for simultaneous determination of six isothiazolinones (MI, CMI, BIT, OIT, DCOIT and MBIT) of water-based adhesive used for food contact materials by optimizing high-performance liquid chromatography/mass spectrometry (HPLC/MS). The main experimental parameters affecting extraction efficiency, such as extraction method, extraction solvent, extraction time and solid: liquid ratio will be optimized by means of single-factor experimental design. The method is validated and can be applied to water-based adhesive used for food contact materials.

## 2. Results and Discussion

### 2.1. Optimization of Extraction of Isothiazolinones in Adhesives

Due to very different polarities of these isothiazolinones, sample preparation, and chromatographic separation are crucial for establishing a suitable method to determine multiple compounds. Four samples containing four kinds of isothiazolinones (MI, CMI, BIT, and MBIT) were chosen for the optimization of pretreatment conditions. The main factors affecting extraction efficiency, such as extraction method, extraction time, extraction solvent and solid: liquid ratio were optimized by single-factor experimental design.

Either ultrasonic extraction or vortexes extraction [32] was commonly used for extraction of organic compounds, with the advantage of easy operation and good extraction capability. The extraction efficiency of above methods was studied and compared by following experimental procedure: 0.50 g of water-based adhesive was weighted accurately and subject to extraction with 20 mL methanol by ultrasonic with 300 W power and vortexed at 1600 r/min for 60 min, respectively. It can be seen from Figure 2a that no significant differences were observed between the concentration of isothiazolinones obtained by applying either ultrasound or vortexes method. Given the practical resources in the lab, the vortexes method was selected to perform the extraction.

The extraction time is a key factor affecting the extraction efficiency as longer time usually increase the kinetic energy of the molecules. In this study, the extraction efficiency under different extraction time (15, 30, 45, 60, and 90 min) for adhesive sample 1, 2, 3, and 4 were investigated. 0.50 g of water-based adhesive sample was weighted accurately and subject to vortexes extraction with 20 mL methanol at different extraction time (15, 30, 45, 60, and 90 min). The results are shown in Figure 2b, which revealed that the extraction time actually had key impact on the concentration of the analytes measured. After 60 min’s treatment, the concentration of analytes maintained stable for most samples measured. Thus, 60 min was determined as the extraction time in vortexes method. It can be seen from Figure 2b that the concentration of BIT, MI and CMI detected in some samples decline after 90 min extraction, indicating that the degradation of isothiazolinones may occur under long time extraction.

The extraction efficiency for different solvents (methanol, water, acetonitrile and methanol-water (1:1)) was studied and compared by following experimental procedure: 0.50 g of water-based adhesive sample was weighted accurately and subject to vortexes extraction with 20 mL of different solvents (methanol, water and methanol-water (1:1)) for 60 min. The results shown in Figure 2c indicated that either methanol or water has the strong extraction capability. The extraction efficiency on BIT, MI, CMI and MIBT were higher by using methanol as solvent than that using water or methanol-water (1:1) mixture. Therefore, methanol was chose as the solvent to perform the extraction of isothiazolinones from water-based adhesive.

The effects of solid:liquid ratio (1:20, 1:40, 1:60, 1:80 and 1:100) on extraction efficiency were also investigated here. 0.2 g, 0.25 g, 0.33 g, 0.5 g, 1.0 g water-based adhesive was accurately weighted, respectively, and subject to vortexes extraction with 20 mL methanol for 60 min. The results shown in Figure 2d suggesting that the extraction efficiency get boosted with the increasing solid:liquid ratio, it can be explained that the solution easily reaches saturation when the solid:liquid ratio is low, and the targeted compounds cannot be extracted more efficiently. In the case that 1:100 of solid:liquid ratio was applied, a stable concentration of analyte was obtained. Thus, 1:100 of solid:liquid ratio was chose for further extraction process.

The optimized extraction condition could be summarized as: 0.2 g water-based adhesive was weighted accurately, and subject to vortexes extraction with 20 mL methanol at 1600 rpm/min for 60 min.

### 2.2. Optimization of Chromatographic and Mass Spectrometric Conditions

Mobile phase system of methanol-water and methanol-water (0.1% formic acid) was evaluated for the selection of mobile phase (See the Appendix A). The results showed that the targeted compounds can be completely and effectively separated by using either methanol-water or methanol-0.1% formic acid. Taking into account that the latter mobile phase has a higher response value and the ESI+ mode was used, the component of the molecule ion is more easily ionized under acidic conditions, thus, methanol-0.1% formic acid was chose as the mobile phase for this experiment.

The effects of column temperature on separation of analytes at 30 °C, 40 °C and 50 °C were investigated in this experiment. With the increase of temperature, the retention time for each analyte was expected to be observed earlier. The compound can be separated well under above three column temperatures. The peak area observed was little change and the influence of column temperature on the targeted analytes is not significant. Therefore, the column temperature was set to 30 °C. (Figure 3a).

The total ion chromatograms (TIC) chromatograms of isothiazolinones in the standard solution and water-based adhesive samples obtained under the above optimized experimental conditions was shown in Figure 3. Owing to the presence of alkyl groups, there are more signals response values observed for the fragment ions of OIT and DCOIT than that of other compounds (See Figure 3b).

Six targeted analytes were injected into the mass spectrometer to determine the characteristic ions. Six isothiazolinones contain a tertiary nitrogen atom in the molecular structure, and a high response value for [M + H]+ molecular ion peak was obtained in the ESI+ mode. The characteristic secondary fragment ions can be consulted on molecular ion under the action of different collision energy in daughter scan mode. The typical daughter scan mass spectrums and MSscan mass spectrums are shown in Figure 4. It can be seen from Figure 4 that the [M + H]+ ion of MI is at m/z 115.8 and the main fragments are 79, 85, 101;the [M + H]+ ion of CMI is at *m/z* 150 and the main fragments are 87, 115, 135; the [M + H]+ ion of BIT is at *m/z* 152 corresponds and the main fragments are 77, 105, 109, 134; the [M + H]+ ion of MBIT is at *m*/*z* 166 and the main fragments are 109, 123, 151; the [M + H]+ ion of OIT is at *m*/*z* 282.5 and the main fragment is 170 and 77; the [M + H]+ ion of DCOIT is at *m*/*z* 214 and [M + Na]+ ion is at *m*/*z* 236 and the main fragment is 101.9 and 57. The proposed fragmentation pattern of MI, CMI, BIT, MBIT, OIT, and DCOIT were speculated in Figure 5. Due to isothiazolinones contain a tertiary nitrogen atom in the molecular structure, the fragments were easier to loss the methyl radical, sulfur negative ion, oxygen negative ion, carbonyl group, amino group, and alkyl groups. The multi-reaction monitoring (MRM) mode was applied to collect the secondary fragments produced by the molecular ion in the ESI+ mode. The quantitative ions of six isothiazolinones (MI, CMI, BIT, OIT, DCOIT and MBIT) were selected according to the daughter fragment ions. The mass spectrometry parameters were optimized to determine the highest responding value of each analyte. Mass spectrometry parameters such as characteristic ions, cone voltage, and collision voltage for the targeted six analytes are shown in Table 4. The MRM chromatogram is illustrated in Figure 6.

### 2.3. Limits of Detection (LOD), Limits of Quantification (LOQ) and Calibration Curves

Under the optimized chromatographic conditions, the calibration curve for each analyte was constructed by plotting the peak area (Y) against the concentration (X) prepared by diluting the standard working solution. The linearity range of MI, CMI, BIT, MBIT, OIT and DCOIT is 0,010–1.0 mg/L, 0,010–1.0 mg/L, 0,010–1.0 mg/L, 0.0025–2.5 mg/L, 0.002–2 mg/L and 0.005–5 mg/L, respectively. A satisfactory linearity was observed. The correlation coefficient for each analyte was R2 ≥ 0.995. Limits of detection (LOD) and quantification (LOQ) were identified as a signal-to-noise ratio of 3 (S/N = 3) and 10 (S/N = 10), respectively. The detection limit (LOD) for MI, CMI, BIT, MBIT, OIT, and DCOIT was determined as 0.01 mg/L, 0.01 mg/L, 0.01 mg/L, 0.01 mg/L, 0.0025 mg/L, 0.003 mg/L and 0.005 mg/L, respectively. The limit of quantitation (LOQ) for MI, CMI, BIT, MBIT, OIT, and DCOIT was determined as 0.02 mg/L, 0.02 mg/L, 0.02 mg/L, 0.005 mg/L, 0.004 mg/L and 0.01 mg/L, respectively.

The data are summarized in Table 1, which demonstrated that the method has satisfactory accuracy, precision, and LOQ. It is well suitable for quantitatively determination of multiple isothiazolinones in the water-based adhesives used for food contact materials.

### 2.4. Accuracy, Precision, and Recovery

Based on the optimized conditions, the accuracy of the method was validated by measuring the recovery of different blank samples spiked with the six isothiazolinones at three distinct levels (0.01 mg/kg,0.5 mg/kg and 10 mg/kg). The precision of the method was validated by the experiment that the samples spiked at each level was detected in six replicated times. The datum is summarized in Table 2. The relative standard deviations (RSD) ranged from 0.9%~5.9%, indicating that the precision and accuracy of the method meet the criteria of methodology.

### 2.5. Test Result for Water-Based Adhesive

The validated method was used to determine the concentration of isothiazolinones in four commercial water-based adhesives used for food contact materials. The results are shown in Table 3. Four isothiazolinones (MI, CMI, BIT and MBIT) were detected with concentration ranging from 2.27 to 23.5 mg/kg, while OIT and DCOIT were not detectable in all samples. MBIT was detected in one water-based adhesive with concentration of 50.17 mg/kg. To our knowledge, it was the first time that MBIT was observed in food contact adhesive. To date, as a new isothiazolinones, MBIT was not authorized by Chinese regulation to be used for food contact adhesive or other food contact materials, therefore no conclusions can be drawn on its potential health risks, and a further investigation will be needed.

## 3. Materials and Methods

### 3.1. Reagents and Standards

Purified water was obtained with a Milli-Q Q-POD^®^ system (Merck KGaA, Darmstadt, Germany); HPLC-grade methanol was obtained from Thermo Fisher (Waltham, MA, USA); MI, OIT and BIT were purchased from Dr Ehrenstorfer (LGC, London, UK); CMI was purchased from BePure^®^; DCOIT was purchased from Anpel (Shanghai, China); MBIT was purchased from Aikang (Jiangsu, China); Acetic acid and ethanol were purchased from Guangshi (Guangzhou, China).

The standard stock solution (1000 mg/L) for each target analyte was prepared by weighting 10 mg of MI, CMI, BIT, OIT, DCOIT and MBIT (accurate to 0.1 mg) respectively, followed by diluted to 10 mL volume with methanol. The intermediate standard solution (10 mg/L) was prepared by transferring 0.1 mL standard stock solution into 10 mL volumetric flasks, followed by diluted with methanol. Working mix standard solutions contained nine targeted analytes (with concentration of 0.01, 0.02, 0.05, 0.1, 0.2, 0.5, 1, 2, 5 mg/L) were prepared by mixing appropriate volumes of the corresponding intermediate standard solution and dilution with methanol.

### 3.2. Sample for Experimental

For determination of isothiazolinones concentration, four different water-based adhesives (Adhesive sample 1, 2, 3 and 4) were provided by various companies for the study. The details for the name and main characteristics of adhesive collected are confidential and cannot be revealed here.

### 3.3. Sample Preparation

For determination of isothiazolinones concentration, weight 0.2 g adhesive sample (accurately to 0.1 mg) and transferred it to the tube, 20 mL methanol was added and subject to vortex extraction for 60 min, then the fraction collected was filtered through 0.22 µm, followed by LC–MS/MS analysis. Each sample was prepared in duplicate.

### 3.4. HPLC–MS/MS Analysis

Target compounds were analyzed by high-performance liquid chromatography (Agilent 1200 series HPLC; Agilent Technologies, Santa Clara, CA, USA) coupled with mass spectrometry (6460 MS; Agilent Technologies, Santa Clara, CA, USA). HPLC separation system was made on a C18 column (Proshell120 EC-C18; 3.0 mm × 150 mm, 2.7 μm) from Agilent Technologies, (Santa Clara, CA, USA).

The mobile phase through gradient elution was prepared by 0.1% formic acid in water as mobile phase A, and methanol as mobile phase B. The retention time, peak width, and resolution of each solute under different mobile phase conditions and column temperature were studied. A optimized HPLC condition was described as follows: the mobile phase was initially started at 50%(B)(0~1 min), it was gradient up to 95%(B) (1~1.5 min) and maintain for 6min, then it was set to 50% at 6.1min, then 50%(B) was kept at 14 min. The flow rate was kept at 0.3 mL/min. The column temperature was kept at 30 °C and the injection volume was 2 µL. (see the Appendix A)

Targeted analytes were measured with electrospray ionization (ESI) in the positive mode. Nitrogen was used as a de-solvation gas and was operated at a flow rate of 6L/min at 300 °C. The nebulizer pressure was 45 psi, the capillary was set at 4000 V, the sheath gas temperature was 380 °C, the sheath Gas Flow was operated at 11 L/min; Nozzle voltage was set at 500 V. Mass spectrum parameters on the strength of the molecular ion and fragments ion were studied in MS scan mode and daughter mode (see Figure 3 and Figure 4). The quantitative of analytes were performed in MRM mode. The optimized MRM parameters for six targeted analytes are shown in Table 4.

## 4. Conclusions

In this paper, a method for the determination of isothiazolinones in water-based adhesives used for food contact materials by LC–MS/MS was established. The main experimental factors affecting extraction efficiency, such as the extraction method, extraction time, extraction solvent, and solid: liquid ratio have been evaluated and optimized by means of single-factor experimental design, using a real adhesive sample. This method was proved to be an effective and reliable technique, which is suitable for simultaneously detecting multiple isothiazolinones at a low level. A satisfactory linear relationship was obtained in a range of 0.010~1.0 mg/L. The LOD and LOQ was 0.010 mg/L and 0.020 mg/L, respectively. The recoveries ranged from 81.5% to 107.3%, and the relative standard deviation was less than 5.9%. Moreover. This analytical method is also suitable for determination of isothiazolinones used for paints, cosmetics, and paper, etc.

The present results obtained in water-based adhesive samples indicated that these studied adhesives containing a combination of isothiazolinones (BIT, MI, CMI and MBIT) with the concentration ranging from 2.27 to 123.5 mg/kg. MBIT was detected with concentration of 50.17 mg/kg. To our knowledge, it is the first time that MBIT, as a new isothiazolinones derivative, was observed in water-based adhesives. Since MBIT is not a regulated compound used for food contact material in China, a further investigation is needed for its migration to food and the risk to human health.

## Figures and Tables

**Figure 1 molecules-24-03894-f001:**
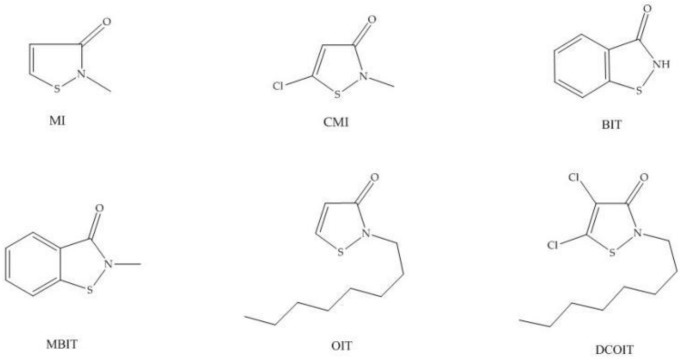
The chemical structures of isothiazolinones.

**Figure 2 molecules-24-03894-f002:**
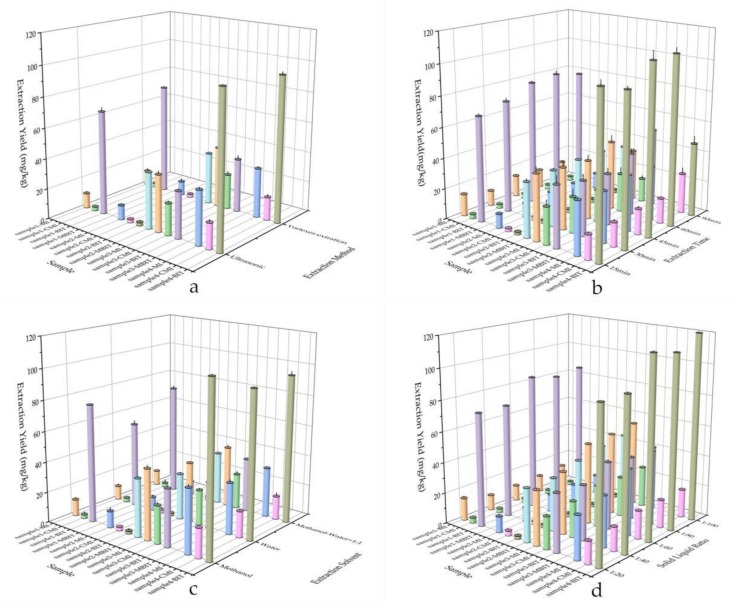
Optimization of extraction of isothiazolinones in adhesives: (**a**) the extraction efficiency of isothiazolinones by different extraction method; (**b**) the extraction efficiency of isothiazolinones by different extraction time in vortexes method; (**c**) the extraction efficiency of isothiazolinones by different solvents; (**d**) the extraction efficiency by different solid: liquid ratio (methanol) in vortexes method. (The standard deviations were used as error bars).

**Figure 3 molecules-24-03894-f003:**
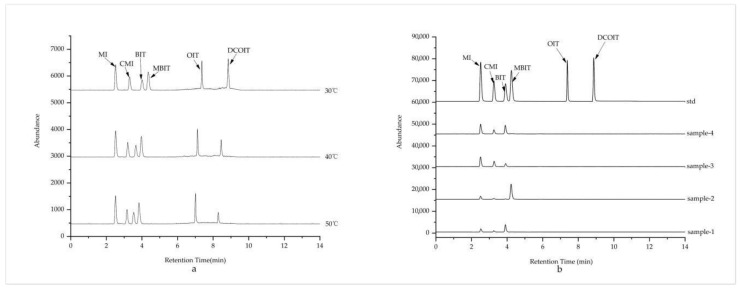
The TIC chromatogram of targeted isothiazolinones: (**a**) The TIC chromatogram of targeted isothiazolinones in standard solution at different column temperatures, (**b**) The TIC chromatogram of targeted isothiazolinones in standard solution and real water-based adhesives in positive ionization mode.

**Figure 4 molecules-24-03894-f004:**
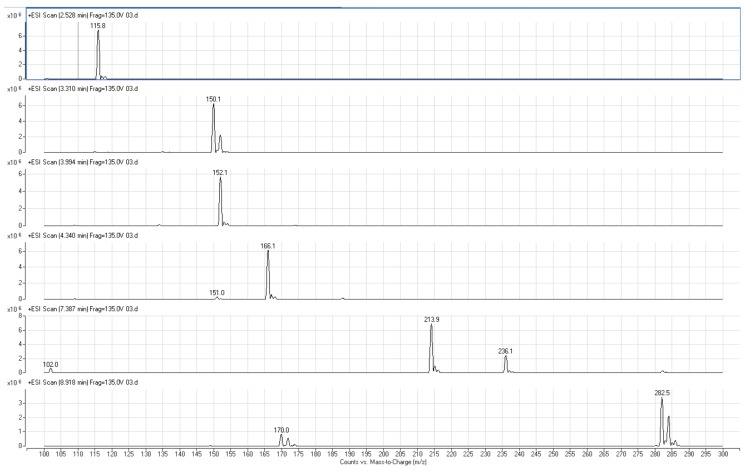
The MS scan and Daughter scan of isothiazolinones.

**Figure 5 molecules-24-03894-f005:**
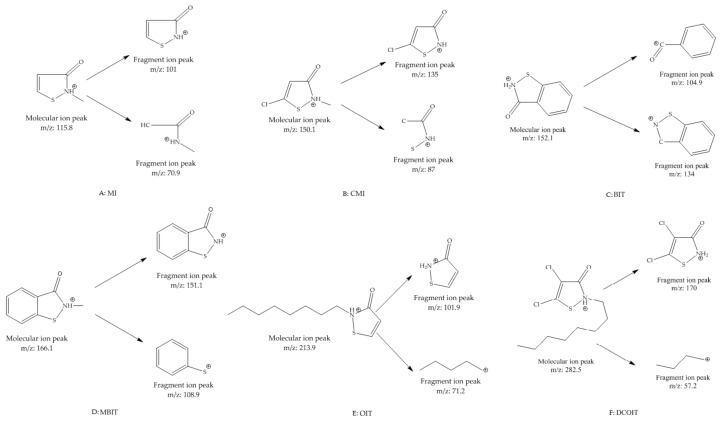
The proposed fragmentation pattern of isothiazolinones.

**Figure 6 molecules-24-03894-f006:**
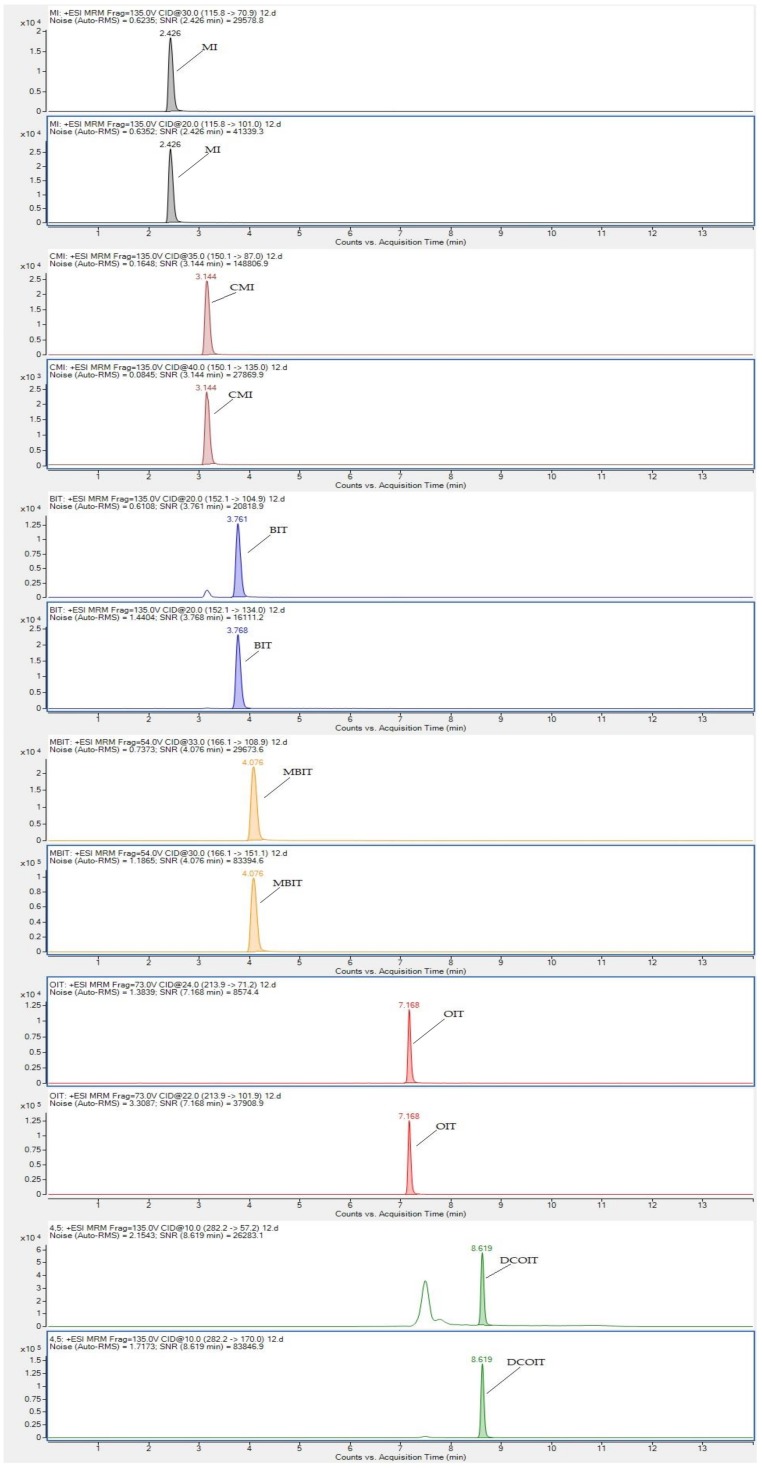
The MRM chromatograms of each targeted isothiazolinone (1.00 mg/kg).

**Table 1 molecules-24-03894-t001:** Calibration curves, limit of detection (LOD) and limit of quantitation (LOQ) for 6 isothiazolinones in the water-based adhesive.

Compound	Linearity Range (mg/L)	Regression Quotation	Correlation Coefficient	LOD (mg/kg)	LOQ (mg/kg)
MI	0.02~10	Y = 110989.2X + 24155.2	0.995	0.01	0.02
CMI	0.02~10	Y = 84968.3X + 28905.5	0.994	0.01	0.02
BIT	0.02~10	Y = 132276.4X + 18210.4	0.995	0.01	0.02
MBIT	0.005~2.5	Y = 700783.1X + 34814.8	0.998	0.0025	0.005
OIT	0.004~2	Y = 440769.4X + 32874.1	0.994	0.002	0.004
DCOIT	0.01~5	Y = 221062.6X + 16943.5	0.997	0.005	0.01

**Table 2 molecules-24-03894-t002:** The recovery and precision for six isothiazolinones spiked in three levels (*n* = 6).

Compound	Spiked (mg/kg)	Detection (mg/kg)	Recovery (%)	RSD (%)
MI	0.01	0.0082	81.5	3.7
	0.5	0.488	97.5	2.1
	10	9.93	99.3	2.5
CMI	0.01	0.0083	83	5.9
	0.5	0.447	89.4	3.6
	10	10.15	101.5	2.1
BIT	0.01	0.0084	84.5	3.1
	0.5	0.446	89.3	4.2
	10	9.23	92.3	3.0
MBIT	0.01	0.0086	86.2	4.3
	0.5	0.490	97.9	4.2
	10	10.27	102.7	0.9
OIT	0.01	0.0090	90.3	3.2
	0.5	0.520	104	1.1
	10	9.72	97.2	3.8
DCOIT	0.01	0.0085	85	3.7
	0.5	0.476	95.3	3.5
	10	10.73	107.3	4.4

**Table 3 molecules-24-03894-t003:** The concentration of six isothiazolinones (mg/kg) in water-based adhesive under optimized experiment condition.

Sample	MI	CMI	BIT	MBIT	OIT	DCOIT
1	13.87	3.61	91.25	ND *	ND *	ND *
2	12.94	3.28	2.27	50.17	ND *	ND *
3	59.21	28.36	45.32	ND *	ND *	ND *
4	60.79	26.73	123.5	ND *	ND *	ND *

* ND refers to non-detectable.

**Table 4 molecules-24-03894-t004:** The optimized MRM parameters for six targeted analytes.

Compound	Precursor Ion/(*m*/*z*)	Product Ion/(m/z)	Dwell Time/ms	Fragmenter/V	Collision Energy/eV
MI	115.8	101.0, 70.9 *	30	135	20, 30
CMI	150.1	135.0, 87.0 *	30	135	40, 35
BIT	152.1	134.0 *, 104.9	30	135	20, 20
MBIT	166.1	151.1 *, 108.9	30	54	30, 33
OIT	213.9	101.9 *, 71.2	30	73	22, 24
DCOIT	282.5	170.0 *, 57.2	30	135	10, 10

* quantitative ions.

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
