# Peer review of "Simultaneous Quantitative Analysis of Six Isothiazolinones in Water-Based Adhesive Used for Food Contact Materials by High-Performance Liquid Chromatography–Tandem Mass Spectrometry (HPLC–MS/MS)"

_molecules, 2019, doi:10.3390/molecules24213894_

Round 1

Reviewer 1 Report

The manuscript presents a method for the determination of isothiazolinones in adhesives intended for food-contact applications. Despite the analysis of such compounds is new, the analytical procedures, sample treatment and optimization are common in most laboratories.

Before its acceptation in "Molecules", major revision must be made in order to reach the quality standards of the journal as follows:

Title: please, delete "optimization of". Most of the research works need an optimization as a part of the whole work, so this word can be deleted without compromising the content of the study.

English must be deeply revised. Numerous sentences along the manuscript contain misspellings and lack of concordance between subject and verb, singular/plural, tenses... and in some cases meaningless sentences are written. Please, consider the full revision of the manuscript by a native speaker.

References 5, 6, 10, 11, 13, 15, 16, 20, 23, 24, 25 are omitted in the text.

The HPLC gradient is a little strange. Normally, the percentage of the more apolar phase (methanol here) increases along the time. In this case, a fast gradient is applied and after 4.5 min it is decreased during the chromatogram, instead after the completion and before the next injection for stabilization of the column. What is the reason?

Table 1: please, provide the same number of decimals for all the data contained into the same column.

Line 135. If acetonitrile is not adequate for HPLC, why authors consider such solvent for extraction? Please, remove completely acetonitrile from the extraction section, since it makes no sense to use a solvent that a priori will be discarded due to known problems of compatibility/interference with the mobile phase comprising methanol.

What are the error bars of figure 1? Standard deviation? Please, explain.

Line 146: "is slightly higher" is not true in all the cases, and taking into account error bars, no significant differences are displayed. Please, modify. Besides, no reference about the extraction time is give. Please, especify.

Authors should mention that the optimization is sequential (only one variable is considered each time, whereas the rest keep constant).

Line 157: according to figure 3 (no error bars are depicted here) results seem to be very similar independently of time. 15 min could be adequate? The graph should be improved, since the differences along the time are limited.

In figure 4, again minimum differences are presented. The addition of error bars could help to decide if they are significant from a statistical point of view or not.

Lines 186-191. A simple description is given, but no data at all are provided.

Table 2. The fitting equations are quite different among analytes. LOD and LOQ are just the same for all the compounds? Probably significant differences are present among them. Please, revise.

Lines 248-251 are redundant with the content of table 3 and could be deleted.

Reviewer 2 Report

Manuscript ID: molecules-598453

Title: Simultaneous quantitative analysis of six isothiazolinones in water-based adhesive used for food contact materials by optimization of high performance liquid chromatography-tandem mass spectrometry (HPLC-MS/MS)

General comments:

The main concern about this research is the aim and importance of the study.

What is the toxicity of isothiazolinones used for different type of industry (e.g. food, cosmetics etc.)? What is the real effect on gastrointestinal system, other systems, skin etc.. “Only” allergy?

There are little information about harmful action of these compounds in the literature data, so what is the point to make such type of the research?

Authors should start their research from toxicity testing of several compounds, estimate their toxic doses/effects at least in vitro, and - as a next step -  the sense is to try detect these compounds in different materials and evaluate if they are in this materials in harmful doses for human health.

Authors must argue and emphasise validity of the study.

What is the applicable value (practical aspect) of the study? For who?

Detailed comments:

Many technical errors are present – there are many unnecessary spaces and vice versa. Manuscript is not strictly edited according to Molecules requirements. English correction by native speaker is needed. Lines 42-44 – the sentence from “The mechanism of action…” has no sense – rephrase it. There should be written additional paragraph in Introduction, which could emphasise the validity of the study comprising harmful health effect of these compounds. It should be also emphasised in the Conclusions. References: only about 12 references are from least 5 years.

Round 2

Reviewer 1 Report

Despite the document has been improved, there are still some aspects that should be revised:

Numerous references appear now as "Error! Reference source not found". Please, revise it and be sure that the correct numbers appear in the final PDF version instead of codes.

Line 59: new isothiazolinones derivative -> new isothiazolinone derivatives

Line 62: in European -> in Europe

Table 1: In the column "product ion", please provide one decimal (even if such number is "0") in all the cases, as suggested the first time. E. G.: 101,70.9* à 101.0, 70.9*

Line 164: vortexer -> vortexed

Lines 188: 033g -> 0.33g

Figure 2: Please, cite textually that the error bars are the standard deviation. Despite commented in the letter, no mention is made in the manuscript.

Table 2: Please, change the lower value of the "linearity range" to the corresponding LOQ instead of the LOD that appears in all the cases.

Reviewer 2 Report

The manuscript was improved and all doubts were explained.